# Concentration Dependencies of Diffusion Permeability of Anion-Exchange Membranes in Sodium Hydrogen Carbonate, Monosodium Phosphate, and Potassium Hydrogen Tartrate Solutions

**DOI:** 10.3390/membranes9120170

**Published:** 2019-12-10

**Authors:** Natalia Pismenskaya, Veronika Sarapulova, Ekaterina Nevakshenova, Natalia Kononenko, Maria Fomenko, Victor Nikonenko

**Affiliations:** Department of Physical Chemistry, Kuban State University, 149 Stavropolskaya st., 350040 Krasnodar, Russia; vsarapulova@gmail.com (E.N.); nevakshenova-ekaterina@yandex.ru (V.S.); kononenk@chem.kubsu.ru (N.K.); mfomenkokubsu@gmail.com (M.F.); v_nikonenko@mail.ru (V.N.)

**Keywords:** anion-exchange membranes, diffusion permeability, anions of carboxylic, phosphoric, tartaric acid, protonation–deprotonation reactions, Donnan exclusion, pore sizes

## Abstract

The concentration dependencies of diffusion permeability of homogeneous (AMX-Sb and AX) and heterogeneous (MA-41 and FTAM-EDI) anion-exchange membranes (AEMs) is obtained in solutions of ampholytes (sodium bicarbonate, NaHCO_3_; monosodium phosphate, NaH_2_PO_4_; and potassium hydrogen tartrate, KHT) and a strong electrolyte (sodium chloride, NaCl). It is established that the diffusion permeability of AEMs increases with dilution of the ampholyte solutions, while it decreases in the case of the strong electrolyte solution. The factors causing the unusual form of concentration dependencies of AEMs in the ampholyte solutions are considered: (1) the enrichment of the internal AEM solution with multiply charged counterions and (2) the increase in the pore size of AEMs with dilution of the external solution. The enrichment of the internal solution of AEMs with multiply charged counterions is caused by the Donnan exclusion of protons, which are the products of protolysis reactions. The increase in the pore size is conditioned by the stretching of the elastic polymer matrix due to the penetration of strongly hydrated anions of carbonic, phosphoric, and tartaric acids into the AEMs.

## 1. Introduction

The extraction of valuable components and nutrients from urban and industrial sewage and liquid animal wastes, as well as from wastewater of food and pharmaceutical industries, is one of the main trends in modern membrane technology [1,2,3]. The combination of biochemical and various membrane methods using electrodialysis (ED) for the final selective extraction and concentration of nutrients, in particular, phosphoric acid anions, leads to a significant cost reduction in fertilizer production, as well as to a decrease in sewage toxicity and volumes [1,4,5]. In addition, ED has shown its usefulness in the extraction of polyprotic carboxylic acids, which are the precursors in biodegradable polymer production, from the products of biochemical processing of biomass [6]; in tartrate stabilization of wine [7]; in peptide fractionation [8]; and in the demineralization of solutions containing amino acids, using, among others, phosphate buffers [9,10].

The development of such technologies is constrained by several challenges. The first of them is related to fouling and destruction of ion-exchange membranes during their long-term operation [11,12,13,14]. The second problem is the relatively low current efficiency per phosphorus or carboxylic acid (amino acid), as well as lower concentration of these substances in ED concentration chambers compared with strong electrolytes, such as NaCl [1,15,16]. It is easy to assume that these challenges are largely caused by the peculiarities of transport of the phosphoric and carboxylic acids anions in the anion-exchange membranes, AEMs. These peculiarities influence the ion-exchange membrane permselectivity, which is mainly studied in relation to solutions of strong electrolytes (NaCl, CaCl_2_, Na_2_SO_4_, etc.). A review of these work is given in [17].

It is known that the permselectivity of membranes is mainly determined by their electrical conductivity and diffusion permeability [18,19,20,21]. It was shown in [22,23,24] that the electrical conductivity of anion-exchange membranes (AEM) and cation-exchange membranes (CEM) can remain almost the same or grow with diluting solutions of phosphoric, carboxylic, or amino acid salts. At the same time, information on the diffusion permeability of AEMs in such solutions is rather scarce. We would like to single out the study by Melnikov et al. [24] from the other studies on this subject. The study showed that the diffusion permeability of CEMs in solutions of carboxylic acid (acetic, succinic, or citric) salts was reduced compared with a solution of NaCl. The reason for this membrane behavior is the low diffusion coefficients of the large anions of these acids, which are co-ions in this case. The diffusion permeability of AEMs in moderately concentrated solutions of acetic, succinic, or citric acid salts, where the anions of these acids act as counterions, is comparable to that measured in NaCl solutions [24]; however, diluting these solutions does not cause a noticeable decrease in the diffusion permeability of AEMs, which is always observed in solutions of strong electrolytes [25,26]. The transport of amino acids is studied better. It is known that, if the CEM, is in the H^+^ form or the AEM is in the OH^–^ form, the zwitterionic forms of the amino acid acquire an electric charge opposite to the charge of the fixed groups of the membrane and are easily transported through the membrane as counterions [27]. This phenomenon is called facilitated diffusion [28] or assisted electromigration [29] of amino acids. The integral diffusion permeability coefficient of AEMs increases with diluting the external amino acid solution because amino acid anions, which are transported by the mechanism of facilitated diffusion, take a bigger portion in the total flux of the diffusing substance [30,31].

In this paper, we analyze the concentration dependencies of the integral coefficient of diffusion permeability of homogeneous and heterogeneous AEMs in solutions of potassium hydrotartrate (KHT) and sodium salts of carbonic and phosphoric acids (NaHCO_3_ and NaH_2_PO_4_) and compare them with similar dependencies obtained in solutions of sodium chloride (NaCl) and discuss the reasons for the different behavior of membranes in solutions of strong electrolytes and ampholytes.

## 2. Materials and Methods

### 2.1. Membranes and Solutions

The homogeneous (AX and AMX-Sb) and heterogeneous (MA-41 and FTAM-EDI) anion-exchange membranes are studied. Some of their characteristics are shown in Table 1. The homogeneous membranes (commercial AMX-Sb manufactured by Astom, Japan [32], and experimental AX) were prepared by the paste method [33]. The ion-exchange matrix of these membranes consists of polystyrene randomly crosslinked with divinylbenzene [34,35]. Polyvinyl chloride granules with a diameter of about 60 nm form the inert binder [36]. The reinforcing cloth is made of the same material. The heterogeneous anion-exchange membranes MA-41 (manufactured by Shchekinoazot OJSC, Russia [37]) and FTAM-EDI (manufactured by Fumatech, Bietigheim-Bissingen, Germany) are made by hot pressing of the ground anion-exchange resin and inert binder (polyethylene) powder. The membranes are reinforced with a nylon cloth. The anion-exchange resin is a regularly crosslinked copolymer of polystyrene and divinylbenzene [37]. The fixed groups in all the investigated membranes are mainly quaternary ammonium cations.

Prior to conducting the investigation, the membranes underwent the pre-test salt treatment [38] and were equilibrated with a 0.02 M NaCl solution.

The solutions were prepared using deionized water (electrical conductivity at 25 °C is 0.5 μS cm^–1^; pH = 5.7 ± 0.2). Crystalline sodium chloride (NaCl), sodium bicarbonate (NaHCO_3_), monosodium phosphate (NaH_2_PO_4_·2H_2_O), and potassium hydrotartrate (KC_4_H_5_O_6_, hereinafter KHT) of analytical grade (manufactured by Vecton OJSC, Russia) were used. The pH values of the investigated aqueous solutions are summarized in Table 2. Some characteristics of the anions of the studied solutions are presented in Table 3.

### 2.2. Experimental Methods

The ***ion-exchange capacity*** (*Q*) of the membranes was determined by substituting chloride ions with nitrate ions [50].

The ***thickness*** of the membranes (*d*) was measured using the high-precision digital micrometer Filetta, IP54 0–25/0.001 mm.

The ***water content*** of the membranes was determined by the method of hot air drying [38] using the equation: *W = (m_sw_ − m_dry_)/m_dry_*. The masses, *m*, of the air-dried (dry) samples and samples swollen (sw) in the 0.02 M NaCl solution were found using the moisture analyzer, MB25 (Ohaus).

The ***pH of the membrane internal solution*** was evaluated by the color indication method using the ability of anthocyanins to change their structure and color depending on the pH of the medium [51]. The color scale of the indicator was made by adding the color indicator (anthocyanins, 20 mg L^–1^) to various buffer solutions. The color of these solutions and the resin AV-17-2P, which has the same ion-exchange matrix and fixed groups as the studied membranes, was registered at constant luminous intensity (ensured by the constant electric power 14 W ± 5% consumed by the light source) and an optical path length of 225 mm. The resin underwent the prior salt treatment and was in contact with the studied 0.01 M NaCl or KHT solution, which contained the color indicator (2 mg L^−1^), for 48 h. AV-17-2P is a convenient object for the color indication method due to having sufficiently large pores that allow large indicator molecules to enter them. In addition, this resin has a pure white color in the air-dried state and when immersed in transparent solutions, i.e., it does not distort the colors of the indicator with additional shades, as it can happen with the studied membranes.

The ***integral coefficient of diffusion permeability*** (*P*) of the membranes was measured in a two-compartment flow cell. The scheme of the cell and the experimental technique are described in detail in [52]. Prior to the experiments, AEM samples were equilibrated with the studied solutions for 48 h. The measurements for one sample were carried out at different solution concentrations starting from the lowest and finishing with the highest value of concentration.

In a separate experiment, the sample of MA-41, which was in contact with the 0.02 M NaH_2_PO_4_ solution for 150 h, was equilibrated with the 0.05 M NaCl solution. The measurements were carried out in NaCl solutions at a temperature of 25 °C.

The ***volume fraction of the intergel spaces*** (*f*_2_) of the membranes was determined by treating the concentration dependences of membranes electrical conductivity in NaCl solutions [18].

***Standard contact porosimetry*** was used to determine the integral volume of sorbed water per gram of dry membrane (*V*, cm^3^ g^–1^
_dry_) or the number of water molecules per fixed group of AEM (*n_w_*_,_ mol_H2O_ mol^–1^
_f.gr_) as functions of the pore effective radius (*r*, nm) and the binding energy of water (*A*, J mol^–1^). The method consists of measuring the equilibrium curve of the integral volume of sorbed water (or other liquid) of the test sample, which is sandwiched between two reference porous samples with a known pore distribution. Porosimetry curves for reference samples are obtained in independent experiments, for example, by using the mercury porosimetry method. A controlled change in the water content in the samples is carried out by evaporation into a closed volume. The fundamental details of this method are described in [53,54].

The number of moles of water per one mole of fixed groups of dry membrane (*n_w_*) is determined by the equation:(1)$$n_{w}=\frac{V\rho_{H_{2}O}}{18Q},$$
where $\rho_{H_{2}O}$ is the volumetric density of water and *Q* is the ion-exchange capacity of the dry membrane, mmol g^–1^
_dry_.

The binding energy of water is calculated as follows:(2)$$A=2\bar{V}\sigma \;\frac{cos\theta }{r},$$
where $\bar{V}$ is the partial molar volume of water (18 cm^3^ mol^–1^), σ is the surface tension (0.072 N m^−1^ [41]), and θ is the contact angle (cosθ was assumed equal to unity).

Prior to the experiments, the studied samples were immersed in 0.1 M solutions of NaCl, NaHCO_3_, NaH_2_PO_4_, and KHT for 150 h and then properly rinsed with deionized water. The rinsing was repeated until the electrical conductivity and pH of water equilibrated with the samples differed from those of initial deionized water by less than 5%.

## 3. Results

### 3.1. Diffusion Permeability of Membranes in Solutions of Different Electrolytes

Figure 1 shows the concentration dependencies of the integral coefficient of diffusion permeability, *P*, of the studied membranes in solutions of NaCl, NaHCO_3_, NaH_2_PO_4_, and KHT. In the case of the heterogeneous membranes, the values of *P* in all the studied solutions turned out to be higher compared with the homogeneous membranes. Similar patterns were noted in many studies, e.g., [26,38]. Diffusion along the macropores played a decisive role in the observed increase in the diffusion permeability. Such pores exist in heterogeneous membranes (see Section 3.2.2) and are nearly absent in the vast majority of homogeneous membranes [54,55]. Another general pattern was the deviation of concentration dependencies with transitioning from NaCl to other electrolytes. In the case of the NaCl solution, the *P* values decreased with dilution of the external solution. In the case of NaHCO_3_, NaH_2_PO_4_, and KHT, on the contrary, a decrease in the salt concentration in the external solution almost always led to an increase in the diffusion permeability. Moreover, this increase intensified in the following sequence: NaHCO_3_ < NaH_2_PO_4_ < KHT.

The reasons for the increase in the diffusion permeability of the heterogeneous membranes as compared with the homogeneous membranes, as well as the decrease in the diffusion permeability of the membranes with the decrease in the concentration of strong electrolyte (NaCl) solutions are well known. In the framework of the microheterogeneous model [18,19,56], the differential coefficient of diffusion permeability *P** is determined by the diffusion permeability of the membrane gel phase, $\bar{P}$, and the diffusion coefficient of the electrolyte, *D*, located in the intergel spaces.
(3)$$P^{*}={\left\{{\left[f_{1}{\left(\frac{\bar{P}}{\bar{t_{1}}}\right)}^{\alpha }+f_{2}{\left(\frac{D}{t_{1}}\right)}^{\alpha }\right]}^{-1/\alpha }+{\left[f_{1}{\left(\frac{\bar{P}}{\bar{t_{A}}}\right)}^{\alpha }+f_{2}{\left(\frac{D}{t_{A}}\right)}^{\alpha }\right]}^{-1/\alpha }\right\}}^{-1},$$
where α is the structural parameter. Its value depends on and characterizes the disposition of the phases (domains) relative to the transport axis. In the considered membranes, the values of α are close to zero. The transport numbers of ions *i* in the gel phase and in the electrolyte are $\bar{t_{i}}$ and $t_{i}$, respectively. The subscript *i* has the values 1 (counterion) and *A* (co-ion). The gel phase, whose volume fraction in the membrane equals *f*_1_, is a microporous swollen medium. It includes the polymer matrix and the polar fixed groups attached to it, as well as the solution of mobile counterions and, to a lesser extent, co-ions, which compensate the charge of the fixed groups. The concentration of the electrolyte in the intergel spaces, which occupy the central parts of the mesopores, macropores, and various structural defects of the membrane and are filled with the electrically neutral solution, is equal to the concentration of the external solution. The volume fraction of intergel spaces in the membrane equals *f*_2_, *f*_1_
*+ f*_2_
*=* 1.

The diffusion permeability of the gel phase, $\bar{P}$, is mainly determined by the concentration ${\bar{c}}_{A}$ and the diffusion coefficient ${\bar{D}}_{A}$ of co-ions in this phase [56,57].
(4)$$\bar{P}=\left(1-\frac{z_{A}}{z_{1}}\right)\bar{t_{1}}\frac{{\bar{D}}_{A}{\bar{c}}_{A}}{c_{A}},$$
where $z_{1}$ and $z_{A}$ are the charge numbers of counterions and co-ions, respectively (the sign of the charge is considered), and $c_{A}$ is the concentration of co-ions in the external solution. The value of $\bar{t_{1}}$ differs little from 1 in a wide range of concentrations. Therefore, co-ions make the decisive contribution to the value of $\bar{P}$. Theoretical estimates [20,56,58,59] and experimental results [52,58] allow concluding that ${\bar{c}}_{A}/c_{A}$ << 1, and ${\bar{D}}_{A}$ is one or two orders of magnitude smaller than the diffusion coefficient in the external solution. With increasing *f*_2_, the volume fraction of the solution, which is equivalent to the external solution, increases. Thus, the diffusion permeability of heterogeneous membranes, whose *f*_2_ is higher than that of homogeneous membranes (Table 1), increases.

To assess the effect of the external solution concentration on the ion-exchange capacity of the membrane and the effect of charges of co-ions and counterions on the diffusion permeability of the membrane, it is convenient to use the following equation [56,57]:(5)$$\frac{\bar{c_{A}}}{c_{A}}=K_{D}^{\left|z_{A}\right|}{\left(\frac{c}{\bar{Q}}\right)}^{\left|z_{A}/z_{1}\right|},$$
where *K_D_* is the Donnan constant. This equation is deduced from the Donnan relation considering that the concentration of co-ions in the gel phase of the membrane is much lower than the ion-exchange capacity ${\bar{c}}_{A}\ll \bar{Q}$
$\left(\bar{Q}=Q/f_{1}\right)$. This equation holds in relatively dilute solutions.

According to Equation (5), for a given value of $\bar{Q}$, the concentration of co-ions in the gel phase decreases with dilution of the solution. A decrease in this concentration leads to a decrease in the diffusion permeability of the membranes, which is observed in the case of strong electrolyte solutions, in particular, NaCl (Figure 1).

Another parameter that affects the diffusion permeability of the gel phase is the ratio $\left|z_{A}/z_{1}\right|$. Indeed, substituting Equation (5) in Equation (4) yields:(6)$$\bar{P}=\left(1-\frac{z_{A}}{z_{1}}\right)\bar{t_{1}}{\bar{D}}_{A}K_{D}^{\left|z_{A}\right|}{\left(\frac{c}{\bar{Q}}\right)}^{\left|z_{A}/z_{1}\right|}$$
or
(7)$$\bar{P}=2\bar{t_{1}}{\bar{D}}_{A}K_{D}\left(\frac{c}{\bar{Q}}\right),\;\left|z_{1}\right|=\left|z_{A}\right|=1$$
(8)$$\bar{P}=\frac{3}{2}\bar{t_{1}}{\bar{D}}_{A}K_{D}{\left(\frac{c}{\bar{Q}}\right)}^{1/2},\;\left|z_{1}\right|=2,\;\left|z_{A}\right|=1.$$

If the values $\left(1-\frac{z_{A}}{z_{1}}\right)\bar{t_{1}}{\bar{D}}_{A}K_{D}^{\left|z_{A}\right|}$ are close for the studied electrolytes, and $c/\bar{Q}\ll 1$, then ${\bar{P}}_{\left|z_{1}\right|=\left|z_{A}\right|=1}<{\bar{P}}_{\left|z_{1}\right|=2,\left|z_{A}\right|=1}$.

Note, that in all the studied solutions, the electric charges of counterions and co-ions were the same and equaled to unity; however, diluting the external solution decreased the diffusion permeability of the membranes in the case of NaCl and increased the diffusion permeability in the case of NaHCO_3_, NaH_2_PO_4_, and KHT. In the next section, we show that the internal solution of membranes immersed in NaHCO_3_, NaH_2_PO_4_, and KHT solutions is enriched with doubly charged counterions. The fraction of such ions in the membrane grows with dilution of the external solution; therefore, the number of co-ions in AEM rises, and $\bar{P}$ grows.

### 3.2. Phenomena Affecting Diffusion Permeability of Membranes in NaHCO_3_, NaH_2_PO_4_, KHT

#### 3.2.1. Enrichment of the Membrane Internal Solution with Multiply Charged Anions

The main difference of NaHCO_3_, NaH_2_PO_4_, and KHT from NaCl is that these electrolytes contain acid anions that go into protonation/deprotonation chemical reactions with water. Appendix A presents protonation/deprotonation reactions, the negative logarithms of the equilibrium constants of these reactions (*pK*), as well as the distribution of molar fractions of the carbonic, phosphoric, and tartaric acid species depending on pH, which are calculated using these constants.

Figure 2 shows the pH of the internal solution and the molar fractions of doubly charged anions of carbonic, tartaric, and phosphoric acids in an ideal homogeneous anion-exchange membrane (*f*_1_ = 1). They were calculated considering the equilibrium constants of the protonation/deprotonation reactions of water (*K_w_*), the stepwise equilibrium constants (*K*_1_ and *K*_2_) for the corresponding acids, as well as the conditions of electroneutrality and ion-exchange equilibrium in the AEM and external solution. The pH values of external solutions corresponding to the maximum concentration of a singly charged anion were used for the calculations.

The calculations are made in accordance with the following equations:(9)$${\bar{c}}_{HT^{-}}+2{\bar{c}}_{T^{2-}}+{\bar{c}}_{OH^{-}}={\bar{c}}_{K^{+}}+\bar{Q},$$
(10)$${\bar{c}}_{H_{2}T}=c_{H_{2}T},$$
(11)$${\bar{c}}_{HT^{-}}=\frac{c_{HT^{-}}}{K_{HT^{-}}^{OH^{-}}c_{OH^{-}}}{\bar{c}}_{OH^{-}}$$
(12)$${\bar{c}}_{T^{2-}}=\frac{{\bar{K}}_{2}}{{\bar{K}}_{w}K_{HT^{-}}^{OH^{-}}}\frac{c_{HT^{-}}}{c_{OH^{-}}}{\left(c_{OH^{-}}\right)}^{2}.$$

Substituting Equations (11) and (12) into Equation (9) gives the following quadratic equation:(13)$$2\frac{{\bar{K}}_{2}}{{\bar{K}}_{w}K_{HT^{-}}^{OH^{-}}}\frac{c_{HT^{-}}}{c_{OH^{-}}}{\left(c_{OH^{-}}\right)}^{2}+\left(\frac{c_{HT^{-}}}{K_{HT^{-}}^{OH^{-}}c_{OH^{-}}}+1\right)c_{OH^{-}}-\left(\bar{Q}+{\bar{C}}_{K^{+}}\right)=0.$$

The solution of Equation (13) is:(14)$$c_{OH^{-}}=\left(-b+\sqrt{b^{2}+4ac}\right)/2a.$$

Here,
(15)$$a=2\frac{{\bar{K}}_{2}}{{\bar{K}}_{w}K_{HT^{-}}^{OH^{-}}}\frac{c_{HT^{-}}}{c_{OH^{-}}};b=1+\frac{c_{HT^{-}}}{K_{HT^{-}}^{OH^{-}}c_{OH^{-}}};c=\bar{Q}+{\bar{C}}_{K^{+}}.$$

The $K_{j}^{i}$ is the ion-exchange equilibrium constant for *i* and *j* counterions. A bar indicates the membrane phase. The presented equations consider the case of KHT. They are similar for the other electrolytes. The deduction of these equations is described in detail in [23]. The concentrations of all the components of the external solution and its pH are the input parameters.

The calculations in accordance with Equations (9)–(14) demonstrated that the pH of the AEM internal solution was close to the pH of the external solution, when *c* ≈ 1 M (Figure 2). If 0.4 M < *c* < 1.0 M, there was a slow increase in the pH and molar fraction of doubly charged counterions in the AEM with dilution of the external solutions of NaHCO_3_, NaH_2_PO_4_, and KHT. If *c* < 0.5 M, a sharper increase in the pH and molar fraction of doubly charged counterions with dilution of the external solution was observed as expected. Note that the pH and the ratio of the molar fractions of singly and doubly charged ions in the external solution remained almost constant over the entire studied concentration range. Thus, the pH and ratios of singly and doubly charged anions in the external and internal AEM solutions differed from one another, and these differences increased with decreasing electrolyte concentration in the external solution. For example, according to the calculations, in 0.01 M external electrolyte solutions, the pH and molar fraction of doubly charged counterions were respectively equal: 4.60 and 0.00 (NaH_2_PO_4_), 8.30 and 0.00 (NaHCO_3_), and 3.60 and 0.15 (KHT), respectively. At the same time, the pH and molar fraction values inside the AEM were 6.58 and 0.19 (NaH_2_PO_4_), 10.10 and 0.38 (NaHCO_3_), and 4.92 and 0.79 (KHT), respectively. Enrichment of the gel phase with doubly charged counterions led to an increase in the concentration of co-ions in it. Estimations in accordance with Equation (5) showed that the decrease in the concentrations of the external solutions from 0.01 to 0.001 M led to the decrease in the co-ion concentrations in the gel phase by a factor of 100 in the case of NaCl and only by a factor of 25 in the case of KHT. All other conditions being the same, the concentration of co-ions in the gel phase, which was in the form of a doubly charged ion, will be 10 times higher than in the case of singly charged anions. In this case, a 7-fold to 8-fold increase in the diffusion permeability of the gel phase can be expected.

The calculation results were in good agreement with the data obtained from the color indication method (Figure 3). Indeed, the white anion-exchange resin, which consisted of the same ion-exchange material as the studied AEM, became pinkish-gray when it was equilibrated with the 0.01 M KHT solution containing the color indicator (Figure 3b). The solution had a bright red color. The color of the solution was close to the pH of 3.56 on the color scale, while the pH of the resin internal solution according to this scale was close to 7.00. This meant that the concentration of protons in the resin decreased by 2.0 mM compared with the external solution. The measured pH values were slightly higher than the calculated values (Figure 2). The reason for this divergence may be the overestimated values of the Donnan constant taken for the calculations, as well as the fact that the protonation/deprotonation reactions involving fixed groups of ion-exchange materials were not considered in the calculations. The fact that these reactions took place was indicated by a shift in the pH of the resin relative to the pH of the external solution observed in the case of 0.01 M NaCl solution (Figure 3c). According to the color, the pH of the external solution was really close to 6.00, while the pH of the internal resin solution was close to 7.60. This meant that the concentration of protons in the internal resin solution was about 2 × 10^−3^ mM lower than in the external solution. A small number of weakly basic fixed groups in AV-17-2P apparently caused such a slight difference in the pH in the case of NaCl.

Thus, the pH of the internal solution of anion-exchange membranes (and resins) was higher than the pH of the external solution. Diluting the external solution increased these differences. At the same time, the AEM internal solution was enriched with doubly charged counterions, and the molar fraction of these ions at a given concentration of the external solution increased in the following sequence: HPO_4_^2−^ < HCO_3_^2−^ << HT^2−^. This led to the fact that, when diluting the external solution, the co-ion concentration in the membrane gel phase decreased to a lesser extent than in the case of the strong electrolyte. The increase in the concentration of co-ions in the gel phase of the membrane increased its diffusion permeability. The higher the fraction of doubly charged ions in the gel phase of the membrane, the stronger the manifested effect.

Indeed, with dilution of the external solution, the highest increase in the diffusion permeability of the membranes was observed in the case of KHT (Figure 1). At the same time, the values of *P* in the range of dilute solutions were higher in the case of NaH_2_PO_4_ than in the case of NaHCO_3_, for which our theoretical estimates predicted a higher concentration of doubly charged counterions in the gel phase. The reason for the decrease in the diffusion permeability of the AEM in diluted NaHCO_3_ solution, apparently, was the alkalinity of the AEM internal solution (Figure 2). High pH values of the internal solution lead to deprotonation of secondary and tertiary amines, i.e., to a partial loss of the ion-exchange capacity of the membranes [60]. The Donnan exclusion of protons, which depends on the concentration of fixed groups, is weakened [59] and, consequently, the molar fraction of strongly hydrated, doubly charged CO_3_^2−^ anions is smaller than in calculations that do not take this phenomenon into account. It is known from the literature [61,62] that the content of weakly basic groups in the studied membranes is noticeable. This is evidenced, e.g., by the higher water splitting rate of these commercial membranes compared with the experimental membranes that contain only quaternary ammonium ions [63]. In the case of NaH_2_PO_4_ or KHT solution, the pH of the internal solution of AEM remained acidic or close to neutral in the studied range of concentrations. Thus, the ion-exchange capacity of membranes in such solutions was higher than membranes in the NaHCO_3_ solution.

The increase in the concentration of co-ions in the gel phase was not the only reason for the higher diffusion permeability of membranes in ampholyte-containing solutions. We demonstrate in the next section that the contact of AEMs with strongly hydrated anions of tartaric, phosphoric, and carboxylic acids leads to stretching of the ion-exchange matrix, which reflects in an increase in pore size and, accordingly, an increase in diffusion permeability of the membranes.

#### 3.2.2. Increase in Membrane Pore Size

It is known [59] that substituting weakly hydrated counterions for strongly hydrated counterions leads to an increase in the fraction of bound water and, accordingly, a decrease in the fraction of free water in the internal solution of studied membranes. As a result, the osmotic pressure in pores increases, the elastic polymer matrix stretches, and the radii of pores increase compared with membranes in contact with counterions that are less hydrated. Oxygen as a constituent of the anions of carbonic, phosphoric, and tartaric acids contributes to their higher hydration compared with the chloride ion (Table 3). Therefore, one can expect an increase in the AEM pore radius when replacing a NaCl solution with NaHCO_3_, NaH_2_PO_4_, and KHT solutions. Figure 4 schematically shows this process.

An indirect evidence of the described effect was the increase in the membrane thickness, *d*, after transferring the AEM from the 0.02 M NaCl solution to KHT and NaH_2_PO_4_ solutions of the same concentration (Figure 5).

Electrostatic repulsive forces between neighbouring fixed groups contribute to the stretching of the elastic matrix and compensate for its liability to compress [59]. Crosslinks of the polymer matrix, on the contrary, counteract this stretching [59]. The AMX-Sb and MA-41 membranes are characterized by the close values of the ion-exchange capacity (Table 1). The difference in their structure is that the ion-exchange resin that the MA-41 membrane is made of is the regularly crosslinked polymer [37]. This regularity, apparently, provides a higher stability to the structure of the polymer matrix than the randomly crosslinked AMX-Sb polymer [34,35]. Therefore, the thickness of MA-41 increased by only 1.8% (KHT) and 3.6% (NaH_2_PO_4_), while the thickness of AMX-Sb increased by 7.6% (KHT) and 8.4% (NaH_2_PO_4_) relative to the NaCl solution. In the case of MA-41, which has the regularly crosslinked matrix, the value of the thickness stopped changing after about 50 h. In the case of AMX-Sb, which is composed of the randomly crosslinked polymer, the growth of the thickness did not stop during the entire observation period. Apparently, prolonged contact with strongly hydrated counterions leads to the rupture of some –C–C– bonds in the most weakly crosslinked parts of the matrix of this membrane. This causes an increase in the swelling of AMX-Sb and similar membranes up to their destruction, observed, for example, during prolonged electrodialysis of liquid media of the food industry, which contain highly hydrated substances [11].

More detailed information on the behavior of the membranes in solutions of different electrolytes was provided by the results of standard contact porosimetry and are presented in Figure 6 and Figure 7.

The effective radii of the pores that dominate the studied membranes were found from the values recorded at the inflection points of the curves presented in *V*–log*r* coordinates. The distribution of water over the ranges of the pore effective radii (Δ*V*) can be estimated from the difference of the values of *V* in the sections of the curve plateau that follow the inflection points. The total water content in the test sample (*V_t_*) was found by the difference in weight of a completely swollen and completely dry membrane. Table 4 and Table 5 summarize the evaluation of the total water content, *V_t_*, in the studied samples, as well as the distribution of water in micro-, meso-, and macro-pores. These results were obtained by processing the curves presented in Figure 6 using the method described in [54].

The data obtained allows concluding that the pores with sizes of about 2, 13, and 30 nm dominate in both the studied membranes. As is known, such pores are typical of ion-exchange materials. The main proportion of water was contained in micro- and meso-pores up to 1 nm in size and from 2 to 13 nm in size (Table 4 and Table 5). Apparently, the number of such pores far exceeded the number of pores with other effective radii. The AMX-Sb membrane also contained pores with effective radii of 60–83 nm. These pores were most likely structural defects; however, the contribution of such pores to the total water content was small. The MA-41 membrane had pores with effective radii of about 100–3000 nm. The first of these pores formed at the areas where the particles of the ion-exchange resin contacted the inert binder (polyethylene). The largest pores were localized at the surfaces of the reinforcing material fibers. The reason for their appearance could be attributed to poor adhesion of the reinforcing cloth, inert binder, and ion-exchange resin. The presence of these pores was confirmed by studying the swollen samples of heterogeneous membranes using SEM [64] and microcomputed tomography [65]. This issue is discussed in more detail in [55]. More interesting, in terms of the problem under consideration, was the fact that the total water content in the studied membranes increased when transferring them from NaCl solution to NaHCO_3_, NaH_2_PO_4_, or KHT solutions (Figure 6). In both AMX-Sb and MA-41, the largest increase in water content was observed in pores from 1 to 13 nm (Table 4). In addition, in the case of MA-41, this increase was also recorded in the pores that formed at the areas where the particles of the ion-exchange resin contacted the inert binder and the fibers of the reinforcing cloth. Apparently, the expansion of macropores localized at the junctions of the ion-exchange composite and the reinforcing cloth of heterogeneous membranes contributed to their high diffusion permeability compared with homogeneous membranes. Apparently, similar phenomena occur in the AX, FTAM-EDI membranes.

It was difficult to distinguish an electrolyte most likely to increase the water content in the homogeneous AMX-Sb membrane. In the heterogeneous MA-41 membrane, the water content increased in the following sequence: KHT < NaHCO_3_ ≈ NaH_2_PO_4_.

Figure 7 shows that, in the case of the NaCl solution, there were about three (AMX-Sb) or four (MA-41) water molecules per fixed group in the pores whose size did not exceed 1 nm. This difference is in good agreement with previous studies [63] and can be explained by the higher degree of hydrophobicity of the AMX-Sb material than that of MA-41 [54].

According to the literature [54,56], such pores (lgA > 3) contain only bound water. This water is associated with a fixed group (quaternary amine (h_R3_ = 1.7 [60]) and counterions, whose hydration numbers and Gibbs hydration energies are presented in Table 3. It is known that the numerical values of hydration numbers largely depend on the method of their determination; however, the general pattern is clear and can be traced: the hydration numbers of the multiply charged anions of carbonic, phosphoric, and tartaric acids (these exact counterions prevail in the membranes under the conditions of the experiment) are significantly high compared with the chloride ion. Therefore, *n_w_* (lg*A* = 3) increases when passing from NaCl to other studied electrolytes.

A comparison of binding energies of water at a given *n_w_* in pores whose effective radii did not exceed 13 nm (dotted lines parallel to the abscissa in Figure 7) provided the information discussed in detail below. In the case of AMX-Sb and NaCl, the number of moles of water per one fixed group was 7.76, *A* = 197 J mol^−1^. At the same value of *n_w_*, the binding energy of water in other electrolytes increased to about 290 J mol^−1^. In the case of MA-41 and NaCl, *n_w_* = 10.15, *A* = 197 J mol^−1^. At the same value of *n_w_*, the water binding energies became equal to 250 J mol^–1^ (KHT), 400 J mol^−1^ (NaHCO_3_), and 650 J mol^−1^ (NaH_2_PO_4_), i.e., the values of *A* increased in the sequence: NaCl < KHT < NaHCO_3_ < NaH_2_PO_4_. It can be assumed that the amount of bound water in micro- and meso-pores, whose presence contributes to the stretching of the ion-exchange matrix of membranes, also increased in the same sequence. This sequence was very similar to the sequence of the Gibbs hydration energy growth: $G_{Cl^{-}}^{0}$ << $G_{T^{2-}}^{0}$ ≈ $G_{HPO_{4}^{2-}}^{0}$ < $G_{CO_{3}^{2-}}^{0}$, (Table 3), though it was not identical. Apparently, the swelling of the MA-41 membrane in the NaHCO_3_ solution was less than expected due to the phenomenon already discussed in Section 3.2.1. The alkaline medium of the internal solution of the AEM led to the decrease in the ion-exchange capacity of the membranes. As a result, the electrostatic repulsive forces between adjacent fixed AEM groups decreased, and the stretching of the elastic ion-exchange matrix were reduced compared to the case of KHT and NaH_2_PO_4_ solutions, in which the gel phase have acidic or close to neutral pH values.

The fact that, in all the examined ranges of the pore size, the water content (expressed as a percentage) was a little dependent on the type of electrolyte (Table 4 and Table 5) allows concluding that the swelling of the ion-exchange material occurred uniformly.

Thus, substituting a NaCl solution with NaHCO_3_, KHT, and NaH_2_PO_4_ solutions indeed leads to an increase in bound water in micro- and meso-pores. Therefore, the contact of AEM with anions of carbonic, phosphoric, and tartaric acids causes the stretching of the ion-exchange matrix. Since doubly charged ions are more hydrated than singly charged ions, the degree of stretching of the ion-exchange matrix and pore size should increase with dilution of the external solution because the proportion of doubly charged counterions increases with dilution of the external solution (see Section 3.2). The consequence of the increase in the pore volume of the ion-exchange material is an increase in the fraction of the electroneutral solution in the membrane (*f*_2_). Apparently, the swelling process leads to a decrease in adhesion between the ion-exchange material, the inert binder, and the reinforcing cloth of MA-41. As a result, the macropores size of this membrane grows and volume fraction of the intergel spaces, *f*_2_, grows more than in the case of the studied homogeneous membranes.

Indeed, the volume fraction of the intergel spaces, *f*_2_, of the AMX-Sb and MA-41 membranes that were immersed in NaH_2_PO_4_ and KHT solutions for 150 h increased by about 20 and 30%, respectively, compared with *f*_2_ of AEMs that were immersed only in the NaCl solution (Table 1).

In accordance with Equation (3), this increase in *f*_2_ increased the diffusion permeability of membranes not only in solutions of salts of carbonic, phosphoric, and tartaric acids, but also in solutions of NaCl (Figure 8).

## 4. Conclusions

The analysis of concentration dependencies of diffusion permeability of homogeneous (AMX-Sb and AX) and heterogeneous (MA-41 and FTAM-EDI) anion-exchange membranes (AEMs) was carried out in solutions of ampholytes (sodium bicarbonate, NaHCO_3_; monosodium phosphate, NaH_2_PO_4_; and potassium hydrogen tartrate, KHT) and a strong electrolyte (sodium chloride, NaCl).

In the case of NaCl solutions, the diffusion permeability of all the membranes decreased with dilution of the external solution. This well-known fact is explained in the framework of the microheterogeneous model. The main reason for the decrease in the diffusion permeability of AEMs was the decrease in the concentration of co-ions in the gel phase due to the increase in Donnan exclusion with dilution of the external solution.

When diluting ampholyte solutions, the diffusion permeability of the membranes, on the contrary, increased, despite the fact that the concentration of co-ions decreased with dilution of the external solution. This increase intensified in the sequence: NaHCO_3_ < NaH_2_PO_4_ << KHT. The main reasons for this growth were (1) the enrichment of the internal solution of AEM with multiply charged counterions and (2) the increase in the pore size of AEM.

The enrichment of the internal solution of AEMs with multiply charged counterions was caused by the Donnan exclusion of protons, which are the products of protolysis reactions. The results of the calculations and experiments showed that the pH of the internal solution of the AEMs was higher than the pH of the external solution. This difference increased with dilution of the external solution. At the same time, diluting the external solution resulted in enriching the internal solution of AEM with doubly charged counterions. This led to the fact that, with dilution of the external solution, the concentration of co-ions in the gel phase of the membrane decreased to a lesser extent than in the case of the strong electrolyte.

The measured membrane thicknesses, standard contact porosimetry data, as well as the obtained values of the volume fraction of the AEM intergel spaces allowed to conclude that the penetration of strongly hydrated anions of carbonic, phosphoric, and tartaric acids into the AEMs led to the stretching of the ion-exchange matrix in all the studied membranes.

## Figures and Tables

**Figure 1 membranes-09-00170-f001:**
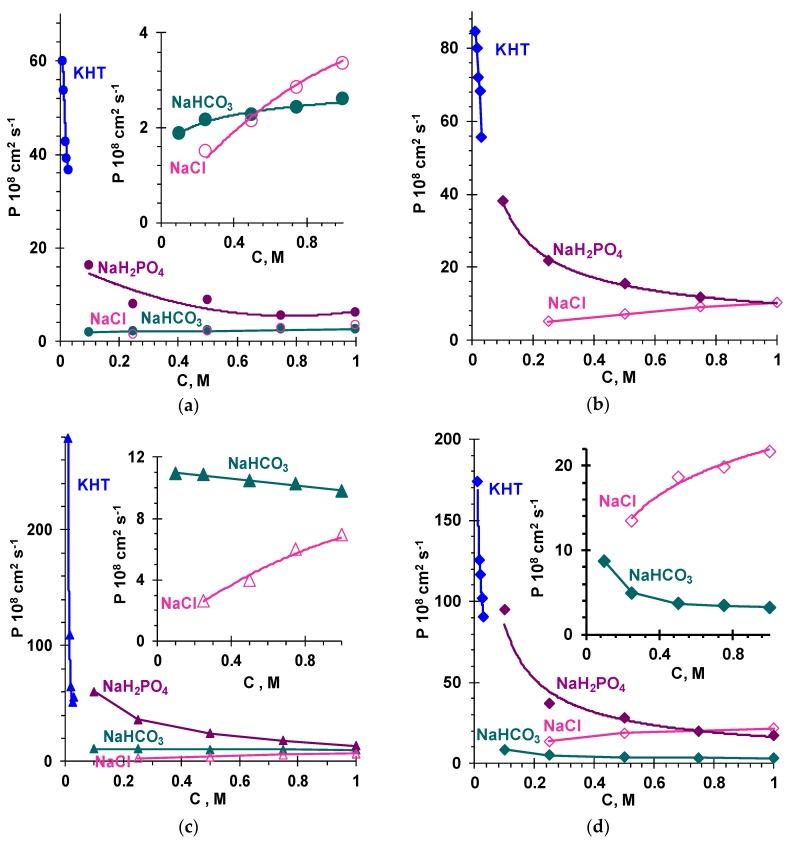
Concentration dependencies of the integral coefficient of diffusion permeability of homogeneous membranes, (**a**) AMX-Sb and (**b**) AX and heterogeneous membranes, (**c**) MA-41 and (**d**) FTAM-EDI in NaCl, NaHCO_3_, NaH_2_PO_4_, and KHT solutions.

**Figure 2 membranes-09-00170-f002:**
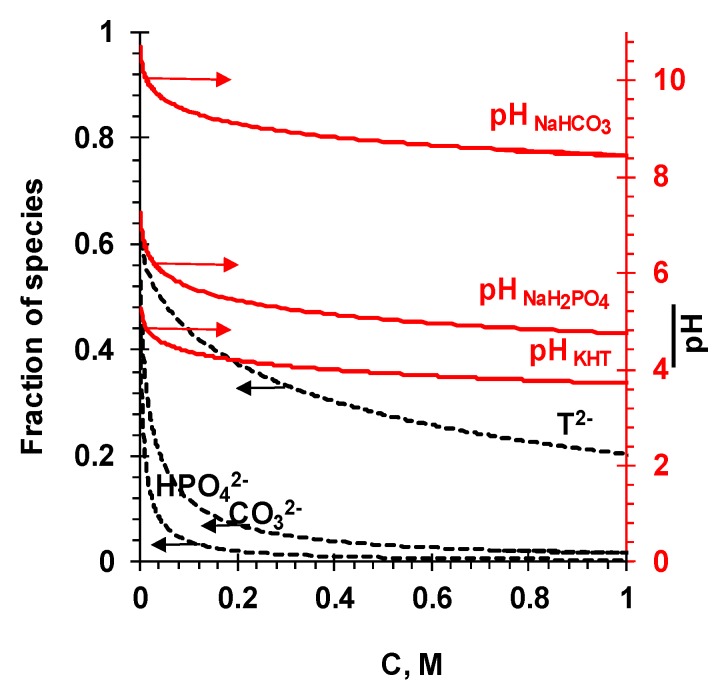
The pH values of the internal solution and the molar fractions of doubly charged anions of carbonic, tartaric, and phosphoric acids in an ideal homogeneous membrane depending on the concentration of the external solution of NaHCO_3_, NaH_2_PO_4_, or KHT. The calculations are made in accordance with the Equations (9)–(14).

**Figure 3 membranes-09-00170-f003:**
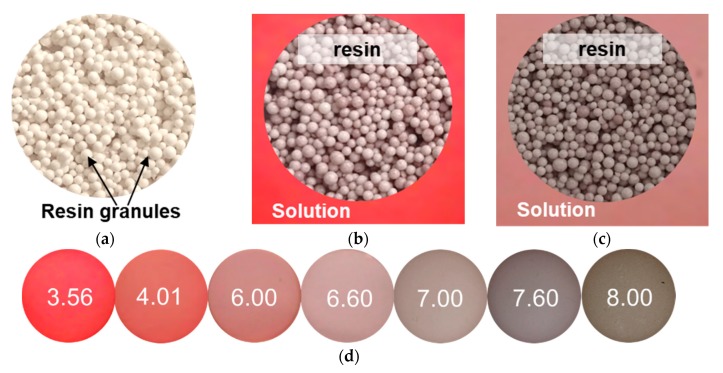
(**a**) The color of the anion-exchange resin AV-17-2P in the air-dried state, and the same resin equilibrated with the solutions of (**b**) 0.01 M KHT, pH 3.7 ± 0.1 and (**c**) 0.01 M NaCl, pH 6.0 ± 0.1 containing the color indicator; (**d**) the corresponding pH of the aqueous solution of the indicator and the color scale of the solution.

**Figure 4 membranes-09-00170-f004:**
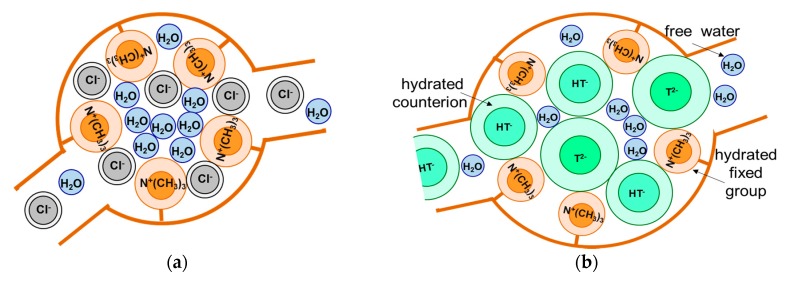
Schematic representation of the effect of hydration degree of (**a**) weakly hydrated (**b**) and strongly hydrated counterions on the size of pores and distribution of free and bound water in them.

**Figure 5 membranes-09-00170-f005:**
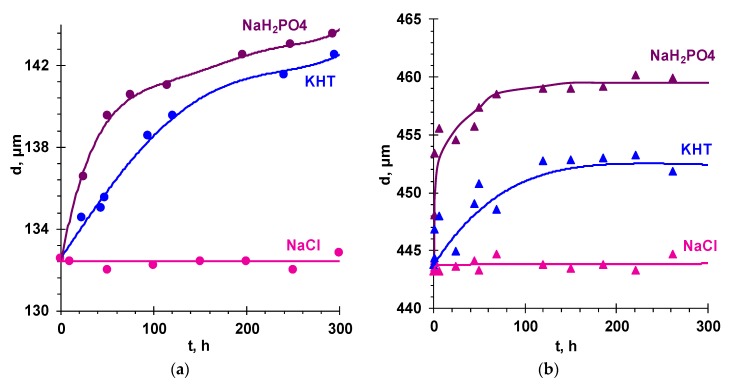
Kinetic dependencies of the thickness of the membranes (**a**) AMX-Sb and (**b**) and MA-41 soaked in 0.02 M NaCl, KHT, and NaH_2_PO_4_ solutions after salt pretreatment and equilibration with the 0.02 M NaCl solution for 48 h.

**Figure 6 membranes-09-00170-f006:**
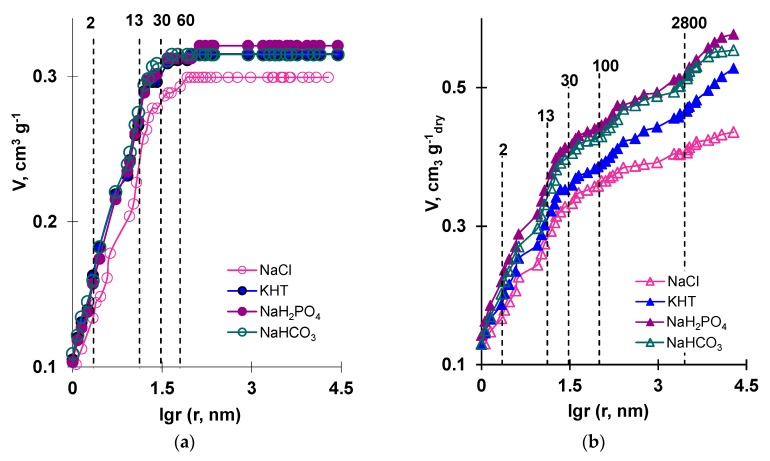
Integral water volume, *V*, distribution vs. the effective pore radii, *r*, in the membranes (**a**) AMX-Sb (**b**) and MA-41. The intersections of the dashed lines with the porosimetric curves indicate the inflection points. These points correspond to the effective radii of the pores dominant in the studied membranes. The numbers above the dashed lines indicate the values of these effective radii.

**Figure 7 membranes-09-00170-f007:**
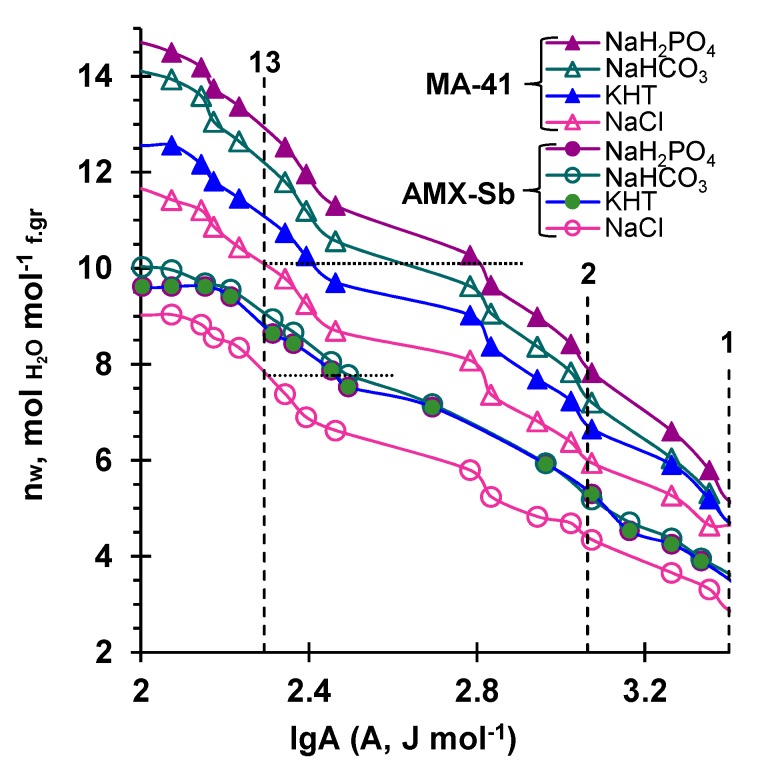
The number of water molecules per one fixed group of a dry membrane, *n*_w_, vs. the binding energy of water, *A.* The data are presented in the pore range typical of the ion-exchange materials of the studied membranes. The dashed lines normal to the abscissa correspond to the binding energies of water in the dominating pores in the case of NaCl solution. The numbers above the curves indicate the effective radii of these pores. The intersections of the dotted lines with the porosimetric curves correspond to the binding energies of water at given values of *n_w_*.

**Figure 8 membranes-09-00170-f008:**
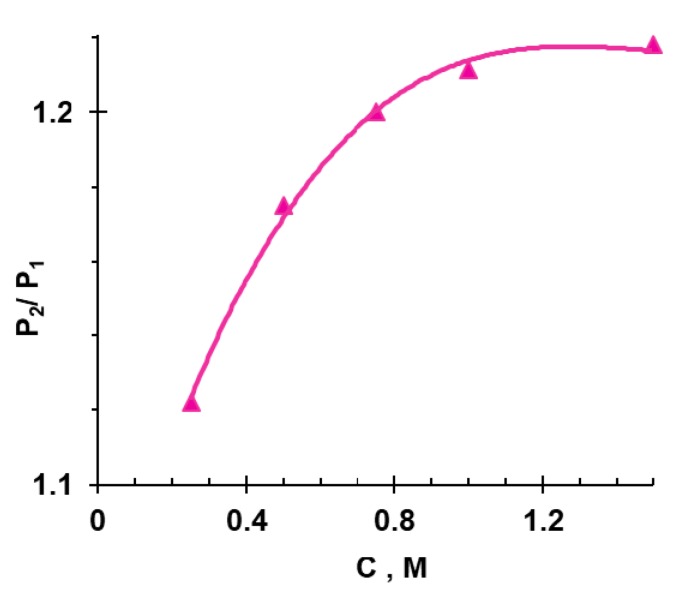
The concentration dependence of the ratio of diffusion permeability of the MA-41 membrane after being in NaH_2_PO_4_ (*P*_2_) and NaCl (*P*_1_) solutions for about 150 h. Data obtained in the NaCl solution.

**Table 1 membranes-09-00170-t001:** Some characteristics of the studied membranes in the swollen (sw) and air-dried (dry) state.

Membrane	Thickness _sw_ ^1^, μm	Water Content _sw_ ^1^, %	Ion-Exchange Capacity, mmol g^–1^ _dry_	Ion-Exchange Capacity, mmol g^–1^ _sw_	Volume Fraction of Intergel Spaces, *f*_2_
AMX-Sb	130 ± 10	22 ± 2	1.71 ± 0.05	1.28 ± 0.05	0.11 ± 0.1
AX	160 ± 5	32 ± 2	2.68 ± 0.05	1.84 ± 0.05	0.12 ± 0.1
MA-41	445 ± 5	28 ± 2	1.56 ± 0.05	1.08 ± 0.05	0.21 ± 0.1
FTAM-EDI	560 ± 5	34 ± 2	1.83 ± 0.05	1.21 ± 0.05	0.15 ± 0.1

^1^ Membrane equilibrated with 0.02 M NaCl solution.

**Table 2 membranes-09-00170-t002:** The pH values of the investigated aqueous solutions.

Electrolyte	pH
NaCl	5.7 ± 0.1
NaHCO_3_	8.3 ± 0.1
NaH_2_PO_4_	4.6 ± 0.1
KHT	3.6 ± 0.1

**Table 3 membranes-09-00170-t003:** Some characteristics of the anions of the investigated solutions.

Counterion	Infinite Dilution Diffusion Coefficients, (T = 298 K), D 10^5^, cm^2^ s^−1^	Stokes Radius (T = 298 K), r_St_, nm [23]	Gibbs Hydration Energy −Δ_hyd_ *G^o^*, kJ mol^−1^	Hydration Number, *h*	
				[39,40]	Other Sources
Cl^−^	2.032 [41]	0.12	340 [39,40]	2.0	
HCO_3_^−^	1.19 [41]	0.21	335 [40]	2.0	5.3 [42]
CO_3_^2−^	0.92 [41]	0.27	1315 [39,40]	4.0	8.5 [42]
H_2_PO_4_^−^	0.958 [41]	0.26	465 [40]	1.5	7 [43] 9 ± 1 [44]
HPO_4_^2−^	0.759 [41]	0.32	1089 [45]	3	10 ± 3 [46]
PO_4_^3−^	0.7 [47]	-	2765 [40]	4.5	13 ± 3 [44] 15 ± 3 [46]
HT^−^	0.852 [41]	0.29	-	-	-
T^2−^	0.805 [23]	0.30	1090 [48]	-	14 ± 3 [49]

**Table 4 membranes-09-00170-t004:** The total water content, *V_t_*_,_ in the AMX-Sb membrane and the distribution of water over the ranges of the pore effective radii, Δ*V*.

Range of the Pore Effective Radii, *r*	Water Content	NaCl	NaHCO_3_	NaH_2_PO_4_	KHT
	Fraction in Total Water Content				
less than 1 nm	ΔV, cm^3^ g^–^^1^	0.089	0.099	0.095	0.094
	%	30	31	30	30
1–2 nm	ΔV, cm^3^ g^–^^1^	0.048	0.061	0.063	0.069
	%	16	19	20	22
2–13 nm	ΔV, cm^3^ g^–^^1^	0.105	0.116	0.112	0.108
	%	35	37	35	34
13–30 nm	ΔV, cm^3^ g^–^^1^	0.043	0.034	0.036	0.036
	%	14	11	11	11
30–60 nm	ΔV, cm^3^ g^–^^1^	0.010	0.005	0.008	0.008
	%	3	2	2	2
60–85 nm	ΔV, cm^3^ g^–^^1^	0.004	0.001	0.004	0.003
	%	2	>1	2	1
	V_t_, cm^3^ g^–^^1^	0.299	0.315	0.321	0.315

**Table 5 membranes-09-00170-t005:** The total water content, *V_t_*_,_ in the MA-41 membrane and the distribution of water over the ranges of the pore effective radii, Δ*V*.

Range of the Pore Effective Radii, *r*	Water Content	NaCl	NaHCO_3_	NaH_2_PO_4_	KHT
	Fraction in Total Water Content				
less than 1 nm	ΔV, cm^3^ g^–^^1^	0.130	0.130	0.142	0.130
	%	30	23	25	25
1–2 nm	ΔV, cm^3^ g^–^^1^	0.038	0.060	0.087	0.060
	%	9	11	15	11
2–13 nm	ΔV, cm^3^ g^–^^1^	0.117	0.156	0.132	0.120
	%	27	28	23	23
13–30 nm	ΔV, cm^3^ g^–^^1^	0.047	0.057	0.054	0.047
	%	11	10	9	9
30–100 nm	ΔV, cm^3^ g^–^^1^	0.027	0.025	0.027	0.027
	%	6	5	5	5
macropores ^1^	ΔV, cm^3^ g^–^^1^	0.077	0.126	0.135	0.144
	%	18	23	23	27
	V_t_, cm^3^ g^–^^1^	0.436	0.554	0.577	0.528

^1^ Localized at the interfaces of the ion-exchange resin particles and the inert material, and at the interfaces of the ion-exchange composite and fibers of the reinforcing cloth.

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
