# Peer review of "Concentration Dependencies of Diffusion Permeability of Anion-Exchange Membranes in Sodium Hydrogen Carbonate, Monosodium Phosphate, and Potassium Hydrogen Tartrate Solutions"

_membranes, 2019, doi:10.3390/membranes9120170_

Round 1

Reviewer 1 Report

The review comments for membranes-664413

The work investigated the changes in diffusion permeability of the anion exchange membranes when running in the system containing pH-sensitive organic anions. It was found that the diffusion permeability is depended on the kind of anion species and their corresponding concentrations. The founding in this work will contribute to the deep observation of the ions transfer in the application-tailored ion exchange membranes, and thus provide effective guidance to industrialization. The work was well-written and fit the readership of membranes. In general, the minor revision was suggested. I have no scientific comments on this work, and only several insignificant questions were addressed below.

A revision on the manuscript title is suggested, i.e., remove or replace the adj “unusual”, a general recognization on “Carbonic, Phosphoric and Tartaric Acids”. Introduction: “We show that the diffusion permeability of AEM increases with diluting solutions of NaH2CO3, NaH2PO4, KHT, while in case of NaCl solutions it decreases. We show that the observed effect is caused by (1) the enrichment of the internal solution of membranes with multiply charged counterions and (2) an increase in the pore size of AEM upon the incorporation of strongly hydrated counterions into them.” Please remove the discussion of the results from the introduction, or place it in the conclusion section. Please check the manuscript carefully to prevent the typo and the display-format errors, i.e., the superscript in chemicals, page 4 line 156, page 6 line 199-200, page 10 line 306, etc. Please refine the discussion to outburst the highlights of the work.

Reviewer 2 Report

Dear authors,

With great pleasure I have read your paper.

The remarks I have can be considered as minor modifications.

First of all, please insert a list of used symbols with their dimension and meaning. This is something we miss right now.

Now I will pass some small errors in the text:

Line 70: Please use NaHCO3 and NOT NaH2CO3

Line 72: Please use NaHCO3 and NOT NaH2CO3

Table 2: Please use subscripts for NaHCO3 and NaH2PO4

Line 199: z1 and zA (I noticed a Russian symbol after z1)

Line 203: After Dappears a Russian symbol (please change)

Line 358: Please explain in a few lines the method as used in reference [54]. The reader can be informed quickly how Table 4 (really important part of the paper) is calculated.  

Fig 6b: Y-axe dimension should read  cm3 gdry-1

Figure 7: Please use for MA-41 : NaHCO3 (and not NaH2CO3)

I will inform the Editor that these minor changes can be revised in a sample manner.

Kind regards,

Reviewer
